# BridgeData V2:
# A Dataset for Robot Learning at Scale

**Homer Walke**[1]   **Kevin Black**[1]   **Abraham Lee**[1]   **Moo Jin Kim**[2]   **Max Du**[2]
**Chongyi Zheng**[4]   **Tony Zhao**[2]   **Philippe Hansen-Estruch**[1]   **Quan Vuong**[3]   **Andre He**[1]
**Vivek Myers**[1]   **Kuan Fang**[1]   **Chelsea Finn**[2]   **Sergey Levine**[1]

[1]UC Berkeley  [2]Stanford  [3]Google DeepMind  [4]CMU

**Abstract:** We introduce BridgeData V2, a large and diverse dataset of robotic manipulation behaviors designed to facilitate research on scalable robot learning. BridgeData V2 contains 60,096 trajectories collected across 24 environments on a publicly available low-cost robot. BridgeData V2 provides extensive task and environment variability, leading to skills that can generalize across environments, domains, and institutions, making the dataset a useful resource for a broad range of researchers. Additionally, the dataset is compatible with a wide variety of open-vocabulary, multi-task learning methods conditioned on goal images or natural language instructions. In our experiments, we train 6 state-of-the-art imitation learning and offline reinforcement learning methods on our dataset, and find that they succeed on a suite of tasks requiring varying amounts of generalization. We also demonstrate that the performance of these methods improves with more data and higher capacity models, and that training on a greater variety of skills leads to improved generalization. By publicly sharing BridgeData V2 and our pre-trained models, we aim to accelerate research in scalable robot learning methods. Project page: https://rail-berkeley.github.io/bridgedata/

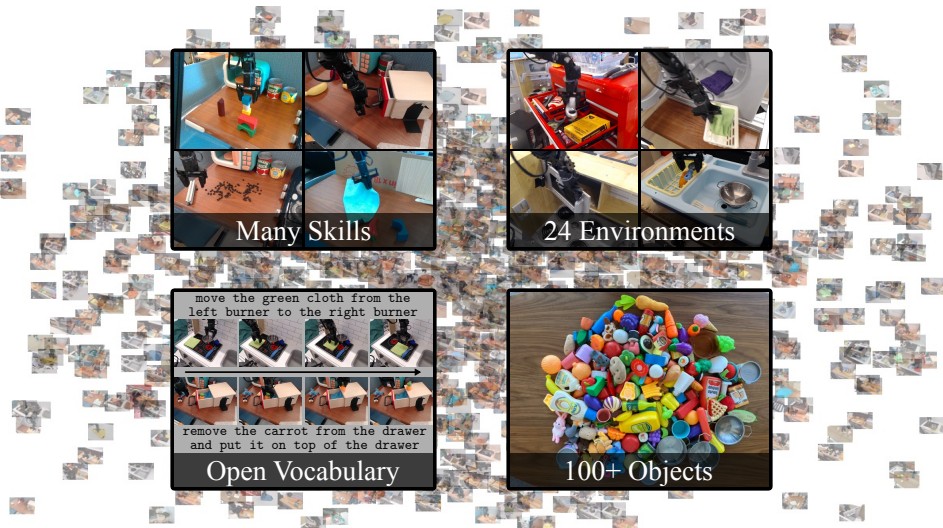

**Figure 1 (BridgeData V2)** We propose a large-scale robotic manipulation dataset containing 60,096 trajectories across 24 environments. The dataset includes skills such as pick-and-place, pushing, sweeping, stacking, folding, and more. Portions of the data include multiple camera views and depth data, and all of the data includes natural language labels.

---

Corresponding email: homer_walke@berkeley.edu

7th Conference on Robot Learning (CoRL 2023), Atlanta, USA.

# 1 Introduction

A useful robotic system needs skills that generalize across the wide variety of conditions found in the real world. Recent results in computer vision and natural language processing have demonstrated that we can obtain broad generalization by training high-capacity models on large and diverse datasets [1, 2]. How can we apply this same recipe in robotics? The general strategy seems straightforward. First, collect a large dataset of demonstrations of robot behaviors. Then, train an expressive policy to extract the desirable behaviors with behavioral cloning or offline reinforcement learning (RL). Given enough data in enough settings, we should learn a policy that can generalize across many tasks and environments.

However, in practice, assembling a dataset with the right features to accelerate research in large-scale robot learning presents a significant challenge. Since collecting a large dataset of robot behaviors is time-consuming, the dataset should be reusable outside of the institution where it was collected. Ideally, any researcher could train on the dataset and obtain a policy with reasonable performance on their tasks. To enable this, the dataset needs extensive coverage of tasks and environments so that policies learned on the data will generalize to new lab settings. Many existing robot datasets contain only one or a few environments and tasks [3, 4, 5], meaning a researcher would need to exactly replicate a scene from the data to use them for robot learning. Additionally, the dataset should support flexible task conditioning, through goal images or natural language instructions, so that researchers can easily command policies trained on the data to perform new tasks. Importantly, the dataset should contain data for many feasible tasks in a given environment so that a multi-task policy must learn to pay attention to the task specification rather than inferring the task from the initial observation.

In this paper, we propose a new dataset, which we call BridgeData V2 (Figure 1) because it greatly expands on the previously released Bridge Dataset [6]. BridgeData V2 contains 50,365 demonstrations of 13 skills across 24 environments, more than 7 times as many demonstrations as the original Bridge Dataset. BridgeData V2 is not just a quantitative improvement: the new dataset is specifically designed as a useful resource for a wide range of robot learning research. The greater quantity and diversity of data enables better generalization to new lab settings and, unlike the first Bridge Dataset, the demonstrations were collected in scenes with many possible skills to support open-vocabulary task specification through either goal images or language instructions. We also augment the demonstrations with 9,731 trajectories collected from a heavily randomized pick-and-place policy to boost the robustness of the foundational objection repositioning skill.

These features make BridgeData V2 well suited for use with a variety of different robot learning methods, many of which make different assumptions. While a number of datasets have been proposed in the past, these have typically only been used with one type of method (e.g., text-conditional behavioral cloning) [5, 7]. To illustrate the versatility and broad applicability of BridgeData V2, we include a comprehensive evaluation of 6 different state-of-the-art robot learning methods, which include algorithms for text-conditioned imitation learning [7], goal-conditioned imitation learning, and goal-conditioned RL [8, 9]. These methods cover a range of key design decisions involving the policy architecture, the use of observation histories, action discretization, and action prediction horizon. We also analyze how the performance of some of these methods changes with dataset size and diversity, showing that large and diverse datasets like BridgeData V2 are an important ingredient in enabling good performance.

Our contributions are a new dataset of robotic manipulation behaviors as well as the empirical study of state-of-the-art offline learning methods using the introduced dataset. We find that BridgeData V2 is useful as an offline dataset for a wide variety of learning algorithms and across multiple labs. Additionally, in our scaling analysis, we find that performance improves with both model size and dataset size and diversity. We believe BridgeData V2 is both a useful resource for robot learning research and a demonstration of the promise of data-driven robot learning.

# 2 Related Work

Prior work on robot learning has often leveraged small datasets that cover a single task in a single domain to develop better learning methods [10, 11, 12, 13, 14, 15, 16, 17, 18]. Several prior projects have assembled large robot datasets for a single behavior like grasping [19, 20, 21, 22, 23], poking [24], pushing [25, 26], or rope manipulation [27]. Others include multiple tasks [3, 4, 28, 29, 30, 31, 5, 32, 33]. However, even the multi-task datasets are difficult for other researchers to use,

| Dataset | # Traj. | # Skills | # Env. | Lang. | Public Data | Public Robot | Collection |
|---------|---------|----------|--------|-------|-------------|--------------|------------|
| MIME [3] | 8.30k | 12 | 1 | ✗ | ✓ | ✓ | human |
| RoboTurk [4] | 2.10k | 2 | 1 | ✗ | ✓ | ✓ | human |
| RoboNet [23] | 162k | n/a | 10 | ✗ | ✓ | ✓ | scripted |
| MT-Opt [30] | 800k | 1 | 1 | ✗ | ✗ | ✓ | scripted & learned |
| BridgeData [6] | 7.20k | 4 | 12 | ✓ | ✓ | ✓ | human |
| BC-Z [5] | 26.0k | 3 | 1 | ✓ | ✓ | ✗ | human |
| RT-1 [7] | 130k | 2 | 3 | ✓ | ✗ | ✗ | human |
| RH20T [40] | 110k | 41 | 50 | ✓ | ✓ | ✓ | human |
| RoboSet [32] | 98.5k | 6 | 11 | ✓ | ✓ | ✓ | 29% human, 71% scripted |
| **BridgeData V2** | 60.1k | 13 | 24 | ✓ | ✓ | ✓ | 84% human, 16% scripted |

**Table 1** BridgeData V2 is a large and diverse publicly available robotic manipulation dataset suitable for a wide variety of learning methods. See Section 3.3 for definitions of "skill" and "environment."

because they are collected in a single environment, and therefore do not facilitate cross-environment generalization. In order for another researcher to use such a dataset, they would need to make their own environment match perfectly. We instead aim to provide a dataset that covers many tasks and many domains, so that other researchers can use the data to experiment with robot learning on large datasets in their own laboratories. Some prior work has proposed large, diverse, and multi-task datasets, but uses proprietary robots [7, 5]. We aim to facilitate academic research by using a low-cost publicly available robot that any lab can acquire. Other projects have tried using low-cost, handheld grasping devices to collect large datasets [34, 35], but this technique requires inferring actions from video and then transferring behaviors to a robot.

Most closely related to this project is the original Bridge Dataset [6, 36]. While our dataset is quantitatively larger, our work also provides a number of qualitative differences (see Table 1). We include both demonstration data and autonomously collected data, as well as provide language descriptions. The aim is to make the dataset useful not only for multi-task imitation learning, but for a variety of learning methods that might have different assumptions: for example, reinforcement learning methods that require data with broader coverage and may benefit from noisy suboptimal trajectories [37], methods that require language descriptions to run language-conditioned imitation learning [5], or methods that use high-capacity models and require very large datasets before they can perform at their full potential [7]. To validate this, we specifically select a wide variety of methods in our evaluation to show that our work provides a single dataset that is compatible with many approaches [7, 38, 39, 17]. This is highly nontrivial, as virtually all prior robotic manipulation datasets have been evaluated with only one or a few methods [23, 5, 7]. Additionally, unlike many prior datasets [23, 5], our experiments isolate the effect of data diversity and show that greater diversity improves generalization, corroborating the result from Brohan et al. [7]. Our experiments extend Brohan et al. [7] by showing that skill diversity improves generalization, not just object diversity. Released earlier this year and collected concurrently with BridgeData V2, RH20T [40] proposes a dataset with several robots and 110k demonstrations. While RH20T is larger, we provide a comprehensive analysis of policies trained on BridgeData V2, and show that these policies can generalize across environments and institutions. Training on a combination of the largest datasets released so far is an exciting and promising direction for future work.

## 3 BridgeData V2

Our goal is to design a dataset that facilitates research in large-scale robot learning. The dataset should support generalization to novel tasks, environments, and even institutions. The dataset should also support flexible task conditioning through either goal images or natural language instructions. In this section, we discuss how we collected the data as well as its composition.

### 3.1 System setup

The robot setup (Figure 2) costs approximately $4,000 in total and consists of parts that are all publicly available with a turnaround time of less than two weeks.

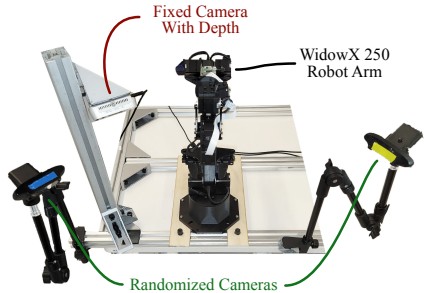

Fixed Camera
With Depth

WidowX 250
Robot Arm

Randomized Cameras

**Figure 2 (System setup)** A picture of our robot setup showing the WidowX 250 robot arm and various cameras. For sensing, we use an RGBD camera that is fixed in an over-the-shoulder view, two RGB cameras with poses that are randomized during data collection, and an RGB camera attached to the robot's wrist. The images are saved at a 640x480 resolution and the control frequency is 5 Hz. We collect demonstrations by teleoperating the robot with a VR controller. See Appendix C for additional details.

### 3.2 Data collection

We designed our data collection protocol to support learning multi-task, generalizable skills. To support broad generalization, we collected data for a wide range of tasks in many environments with suitable variations in objects, camera pose, and workspace positioning. To support the evaluation of multi-task learning methods, we collected data for many possible tasks simultaneously in each environment. This ensures that a policy must pay attention to the task specification, rather than inferring the task from its observations. Notably, this philosophy differs from the original Bridge Dataset, where data was collected for a specific (smaller) set of predefined tasks.

For example, we might set up a kitchen scene with several food items and utensils as well as a drawer. Then, the data collector collects demonstrations by performing any feasible task, such as opening the drawer or putting a utensil in the sink. To speed up data collection, we do not require the positions of the objects in the environment or the robot to be reset between trajectories. We also do not require the data collector to label trajectories with task names. Every 50 trajectories, the collector randomizes the poses of the cameras, switches out the objects in the scene, and randomizes the position of the workspace relative to the robot.

Repositioning objects is a widely applicable skill and thus we wanted to learn policies that can pick-and-place a large number of objects in many environments. During the data collection process, we realized we could partially automate the collection of pick-and-place data with a heavily randomized scripted policy. While this policy fails frequently, we can run it autonomously to collect a large amount of pick-and-place data for a wide range of objects more quickly than teleoperating the robot. Methods that benefit from suboptimal data, such as offline RL, can leverage this autonomous data to learn more robust behaviors. Note that users are free to exclude the autonomous data from training, but our dataset offers the flexibility to explore both options.

Since we do not annotate trajectories with task names during data collection, we used a crowdsourcing platform to label the data post-hoc. Annotators were asked to describe the task being performed by the robot in each trajectory, with particular emphasis on the final location of any moved objects.

### 3.3 Dataset composition

To effectively describe the composition of our dataset, we will first define "skill" and "task," two terms that have had many different meanings in prior work. We use "skill" to denote groups of trajectories that exhibit similar motions — such as pick-and-place, sweeping, folding, or door opening — but potentially with different objects or arrangements of objects. We use "task" to denote groups of trajectories that correspond to a similar language instruction, which in practice typically means that similar motions (e.g., pick-and-place) with different objects (e.g., a fork or a bowl) correspond to the same skill, but to different tasks. This usage of the terms resembles some prior work on language-conditioned robot learning [7, 5].

BridgeData V2 includes 13 skills that range in complexity. A large portion of the data consists of the foundational skills of pick-and-place, pushing, and reorienting objects, since these skills are applicable in a wide variety of settings and mastery of these skills may transfer to more complicated behaviors. The more complex skills include opening and closing doors and drawers, wiping surfaces, folding cloths, stacking blocks, twisting knobs, flipping switches, turning faucets, zipping zippers, and using tools to sweep granular media. To ensure that we can learn generalizable skills, the data includes examples of each these skills applied to a wide variety of objects and in a wide variety of environments. BridgeData V2 features 24 environments, including kitchens, sinks, and tabletops, as well as more than 100 objects. In total, BridgeData V2 contains 50,365 expert demonstrations and

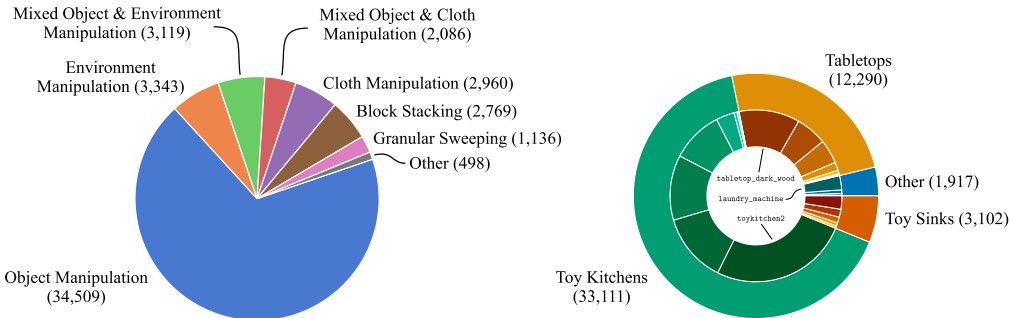

**Figure 3 (Dataset composition) (L)** The various types of tasks in BridgeData V2. The majority of the data comes from foundational object manipulation tasks, such as pick-and-place, pushing, and sweeping. Additional data comes from environment manipulation, which includes opening and closing doors and drawers. The remaining data comes from more complex tasks, such as stacking blocks, folding cloths, and sweeping granular media. Some segments of the data contain mixtures of these categories. **(R)** The 24 environments in BridgeData V2, grouped into 4 categories. The majority of the data comes from 7 distinct toy kitchens, which include some combination of sinks, stoves, and microwaves. The remaining environments come from diverse sources, including various tabletops, standalone toy sinks, a toy laundry machine, and more.

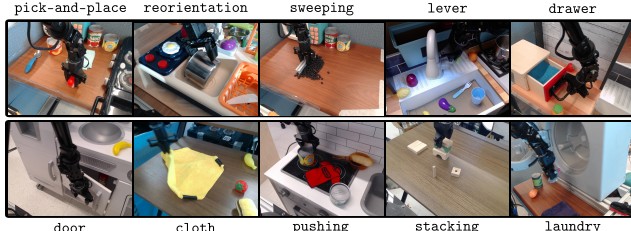

**Figure 4 (Skills + Environments)** Examples of skills and environments in the dataset. BridgeData V2 includes a wide variety of skills, environments, and objects to support broad generalization.

9,731 trajectories from a scripted policy. Figure 3 shows a breakdown of the environments and skills in the demonstration data and Figure 4 shows some examples. We provide additional statistics in Appendix A.

## 4 Offline Learning Methods

To demonstrate that BridgeData V2 is compatible with a variety of learning methods with different assumptions, we evaluated several state-of-the-art offline learning methods using the data. We selected both goal-conditioned methods (in the form of goal images) and language-conditioned methods because our dataset is designed for open-vocabulary task specification. The observation space for all of the methods is 128x128 RGB images, except for RT-1 which uses 320x256 RGB images. We use only the over-the-shoulder camera view. The 7D action space of the robot consists of continuous 6D Cartesian end-effector motion, corresponding to relative changes in pose, as well as a discrete dimension to control the opening and closing of the gripper. We provide the full implementation details and training data statistics in Appendix B.

### 4.1 Goal-conditioned methods

**Behavioral cloning with goals (GCBC).** As a baseline, we evaluate a standard goal-conditioned behavior cloning method. We use a ResNet-34 to encode the observation and goal. The image encoding is then passed through several fully connected layers to produce a robot action.

**Diffusion behavioral cloning (D-GCBC).** Prior work has shown that highly expressive policies can better model multi-modal action distributions, leading to improved performance on a variety of tasks [41, 18, 38]. We modify our GCBC baseline to represent the policy using a diffusion process and train with the DDPM objective [42].

**Action chunking with transformers (ACT).** Previous work has also noted that predicting action sequences instead of single actions can benefit imitation learning [17, 18]. ACT instantiates this with transformers and a conditional VAE [43] training objective, allowing the model to generate multi-modal action sequences.

**Contrastive RL (CRL).** Along with imitation learning methods, we also evaluate a goal-conditioned reinforcement learning method. CRL casts goal-conditioned RL as a representation learning problem, parameterizing goal-conditioned value functions as log-linear representations analogous to contrastive learning [39, 44, 45].

### 4.2 Language-conditioned methods

**Behavioral cloning with language (LCBC).** As a baseline, we evaluate a standard language-conditioned behavior cloning method [46]. The natural language instruction is first encoded using the MUSE sentence embedding [47]. Then the image observation is encoded using a ResNet-34 with FiLM conditioning on the language encoding [48]. The final encoding is then passed into a fully connected policy network to produce a robot action.

**RT-1** [7] is a large transformer model trained via behavior cloning to predict low-level robot actions. The inputs are first tokenized using a pre-trained EfficientNet [49] (for images) and a pre-trained language model [50] (for the language instruction). These tokens are then passed to a decoder-only transformer to predict discretized robot actions. RT-1 is the only method to take a history of observations as input and uses a higher image resolution than the other methods.

## 5 Experiments

The goal of our experiments is to evaluate the utility of our dataset for testing a variety of multi-task offline learning methods. Rather than attempting to rigorously analyze different design decisions in the learning methods, we aim to determine the extent to which our dataset facilitates large-scale robot learning research. Our experiments are designed to answer the following questions:

1. Can we learn a broad range of tasks with both goal and language-conditioned methods?
2. Can we generalize the skills in the dataset to new objects and environments?
3. Can another institution use the dataset to obtain skills that work in their lab?
4. How does model size, dataset size, and dataset diversity affect the performance of our learned policies?

### 5.1 Can we learn a broad range of skills that generalize to new objects and environments?

We first evaluated the methods in Section 4 on tasks that are seen in the training data (see Table 2). Even though these tasks are seen in training, the methods must still generalize to novel object positions, distractor objects, and lighting. These tasks test a range of abilities: opening the drawer requires precisely inserting the gripper into the handle, sweeping the beans involves using a tool and predicting the motion of granular media, stacking a block requires delicate balancing, grasping the corn cob requires correctly orienting the wrist joint, and flipping the pot requires coordinating all six degrees of freedom. To obtain success rates for each method, we collected 10 trials for each task, varying the positions of objects and distractors between trials. Most apparently, RT-1 is significantly better than our LCBC baseline, likely due to a combination of design decisions such as larger images, action discretization, and observation histories. The goal-conditioned methods are comparable to each other in success rate. We also noticed qualitative differences in the robotic behaviors each method produced. For instance, D-GCBC, ACT, and RT-1 exhibit more pauses, because their expressive policy distributions can accurately model pauses in the demonstration data. However, since D-GCBC lacks observation histories like RT-1 or action chunking like ACT, it sometimes produces jerky behaviors indicating oscillation between modes in the action distribution. These behaviors are best observed in the video on our website.

Next, we evaluated the methods on tasks that require generalizing skills in the data to novel objects and environments (see Table 3). For instance, in the rice sweeping task, both the rice and brush are unseen. In the cloth folding task, the cloth is a different color and thicker than the cloths in the training data. In the task of putting a marker into a bowl, the entire environment is unseen, including the objects. The large and diverse training set enabled a considerable degree of generalization, with most methods attaining non-zero success on most tasks. However, the language-conditioned methods particularly struggled on tasks involving unseen objects since these object names are not grounded in the dataset. Once again, RT-1 greatly outperformed the LCBC baseline.

### 5.2 Can another institution use the dataset?

To demonstrate that the dataset is useful for other researchers, we had another institution set up the robot and evaluate the same methods in their lab. The tasks and environments in this evaluation

| Task | GCBC | D-GCBC | ACT | CRL | LCBC | RT-1 |
|---|---|---|---|---|---|---|
| Open drawer | 0.4 | 0.6 | 0.5 | 0.4 | 0.5 | 1.0 |
| Sweep beans into pile with bar | 0.9 | 0.9 | 0.9 | 0.7 | 0.4 | 0.6 |
| Fold thin blue cloth over object | 0.4 | 0.7 | 0.7 | 0.5 | 0.5 | 0.9 |
| Stack green block on yellow block | 0.4 | 0.2 | 0.3 | 0.6 | 0.0 | 0.0 |
| Put corn in pot | 0.9 | 0.8 | 0.8 | 0.8 | 0.0 | 0.0 |
| Put carrot on plate | 0.7 | 0.4 | 0.1 | 0.0 | 0.0 | 0.8 |
| Flip pot upright | 0.1 | 0.1 | 0.0 | 0.4 | 0.4 | 0.4 |
| Put eggplant in pot | 0.1 | 0.2 | 0.0 | 0.0 | 0.0 | 0.2 |
| **Average** | **0.49** | **0.49** | **0.41** | **0.42** | **0.23** | **0.49** |

**Table 2 (Evaluation on seen tasks)** Overall, the goal-conditioned methods were comparable. However, RT-1 outperformed LCBC. Success rates are averaged over 10 trials.

| Task | GCBC | D-GCBC | ACT | CRL | LCBC | RT-1 |
|---|---|---|---|---|---|---|
| Sweep rice into pile with brush* | 0.6 | 0.0 | 0.3 | 0.3 | 0.0 | 0.1 |
| Fold thick gray cloth over object* | 0.3 | 0.6 | 0.7 | 0.0 | 0.0 | 0.4 |
| Put marker in bowl† | 0.6 | 0.6 | 0.2 | 0.7 | 0.0 | 0.0 |
| Wipe the table with the cloth‡ | 0.6 | 0.5 | 0.4 | 0.6 | 0.4 | 0.9 |
| Put the mushroom in the pot‡ | 0.7 | 0.9 | 0.1 | 0.7 | 0.1 | 0.6 |
| Put the spoon on the cloth‡ | 0.8 | 0.7 | 0.0 | 0.8 | 0.0 | 1.0 |
| **Average** | **0.60** | **0.55** | **0.28** | **0.52** | **0.08** | **0.50** |

* Unseen objects, seen environment † Unseen objects, unseen environment
‡ Seen objects, unseen environment

**Table 3 (Evaluation on unseen tasks)** Either the objects, environment, or both objects and environment were unseen. We found that both language and goal conditioned methods exhibit non-zero success, demonstrating that BridgeData V2 supports broad generalization. Success rates are averaged over 10 trials.

matched tasks seen in the training data; however, there were differences in robot setup, camera placement, lighting conditions, and object instances that led to a significant domain shift. We evaluated 3 tasks in our lab (Lab 1) and at the other institution (Lab 2) with all the methods. As seen in Table 4, performance was somewhat worse in Lab 2, but all methods attained non-zero success. RT-1 especially showed only a small degradation in performance. The performance of goal-conditioned methods degraded more, though we note that the "flip the pot upright" task is particularly difficult for a goal-conditioned method since the orientation of the pot is difficult to distinguish. Additionally, the "put eggplant in pot" is a very challenging task in both labs since the eggplant easily slips out of the gripper. Note that these evaluations were performed zero-shot, without any new data collected in Lab 2, and we expect fine-tuning on a small amount of data in a new lab to significantly improve performance, as discussed in prior work [6].

### 5.3 How does varying the size of the model and the size and diversity of the dataset affect performance?

Lastly, we analyzed how the performance of our goal-conditioned BC method varies with model size and dataset size and diversity (see Figure 5). First, we tested GCBC with different sizes of image

| Task | Lab 1 → Lab 2 | | | | | |
|---|---|---|---|---|---|---|
| | GCBC | D-GCBC | ACT | CRL | LCBC | RT-1 |
| Put carrot on plate | 0.7 → 0.3 | 0.4 → 0.0 | 0.1 → 0.0 | 0.0 → 0.3 | 0.0 → 0.0 | 0.8 → 0.4 |
| Flip pot upright | 0.1 → 0.0 | 0.1 → 0.2 | 0.0 → 0.3 | 0.4 → 0.2 | 0.4 → 0.1 | 0.4 → 0.6 |
| Put eggplant in pot | 0.1 → 0.1 | 0.2 → 0.2 | 0.0 → 0.0 | 0.0 → 0.1 | 0.0 → 0.0 | 0.2 → 0.2 |
| **Average** | **0.30 → 0.13** | **0.23 → 0.13** | **0.03 → 0.10** | **0.13 → 0.20** | **0.13 → 0.03** | **0.47 → 0.40** |

**Table 4 (Cross-institution evaluation)** We find that both goal-conditioned and language-conditioned methods achieve non-zero success at a new institution (Lab 2), demonstrating that our dataset is useful for other researchers. Success rates are averaged over 10 trials.

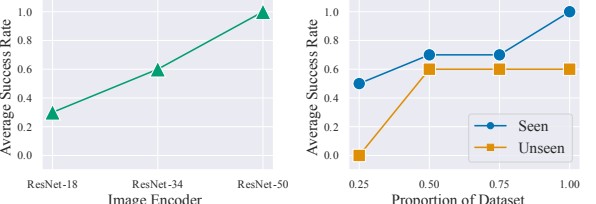

| Train on 3 skills | Train on 13 skills |
| --- | --- |
| 0.30 | **0.65** |

**Figure 5 (Scaling analysis) (L)** Performance of goal-conditioned behavior cloning trained on BridgeData V2 improves with higher capacity models and larger training sets, suggesting that our dataset provides an effective foundation for studying scalable policy learning. **(R)** We compared the performance of GCBC trained on a dataset with only 3 skills and a dataset with all 13 skills, keeping the size of the datasets approximately the same. On an unseen task, the policy trained with greater skill diversity performed significantly better, indicating positive transfer between skills. We used 20 trials per policy in this experiment.

encoders on the task of moving a spoon. We found that higher capacity models perform strictly better at this task. Qualitatively, the smaller models often failed to correctly rotate the gripper to pick up the spoon when it was positioned horizontally. Next, we evaluated GCBC trained on datasets of different size. To create smaller datasets, we randomly subsampled the data stratified by task, preserving diversity while decreasing the number of trajectories. We found that as dataset size increases, performance improves for both a seen and unseen task. To test how skill diversity affects performance, we trained GCBC on two datasets: one dataset with only 3 skills (pick-and-place, pushing, and wiping) and another dataset with all 13 skills. We kept the size of these datasets approximately the same (28k vs 27k trajectories). We found that performance on an unseen pick-and place task was significantly improved by training on data with greater skill diversity. This result indicates that data from other skills can improve the robustness of the pick-and-place skill. Brohan et al. [7] also isolate data diversity and find that greater diversity improves generalization, however they study diversity in objects. We analyze skill diversity and arrive at a similar conclusion. Taken together, the results in this section indicate that scaling the dataset and model capacity further is a promising path toward greater capabilities.

## 6  Discussion, Limitations, and Future Work

We presented BridgeData V2, a dataset with 60,096 trajectories of robotic manipulation behaviors designed to enable research on scalable robot learning methods. Our dataset covers a wide range of tasks and environments and supports methods with various assumptions, including goal and language conditioned imitation learning and reinforcement learning. BridgeData V2 was collected with a low-cost and widely available robot arm, making it suitable for accessible research on robot learning. Our evaluation also demonstrates that a wide range of learning algorithms can use BridgeData V2 to produces policies that generalize across tasks, environments, and institutions. We believe this is a strong validation of the utility of our dataset, as many previously proposed datasets have only been evaluated with one or a small number of methods, and typically at a single institution.

Our dataset does have a number of limitations. While we provide a variety of behaviors, including manipulation of deformable objects, tools, and furniture, the tasks are generally low-precision and do not require complex manipulation of forces. This is reasonable for studying generalization but does not cover the challenges that might occur with more forceful manipulation or more dynamic tasks, such as throwing, moving heavy objects, or low-tolerance industrial insertion. Pushing the frontier on more precise, dexterous, and dynamic tasks is an exciting avenue for future data collection efforts. Additionally, our dataset was collected entirely at a single institution, and although we demonstrate that by utilizing a variety of environments our data enables robots to perform tasks in other labs, an even broader coverage of environments would be a desirable property in future datasets. Finally, although we deliberately chose an accessible and low-cost robot arm, it may be difficult for other researchers to standardize around the same robot. Therefore, a particularly exciting direction for future data collection work would be to assemble multi-robot datasets that enable some degree of generalization across robot morphology. Nonetheless, we hope that our dataset will serve as an important stepping stone toward making scalable robot learning research practical, accessible, and effective.

**Acknowledgements**

We thank Abraham Lee, Mia Galatis, Caroline Johnson, Christian Aviña, Samantha Huang, and Nicholas Lofrese for collecting data, as well as Microsoft Research for assistance in labeling parts of the data with language. This research was supported by the TPU Research Cloud. This research was partly supported by ONR N00014-20-1-2383 and NSF IIS-2150826.

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

| Skill |
| --- |
| Pick-and-place (includes reorienting objects in place) |
| Pushing objects |
| Wiping (e.g., wiping the table with a cloth) |
| Sweeping (e.g., sweeping beans into a pile) |
| Stacking |
| Folding cloths |
| Opening and closing drawers |
| Opening and closing doors |
| Opening and closing cardboard box flaps |
| Twisting knobs |
| Flipping switches |
| Zipping and unzipping |
| Turning levers |

**Table 5** The full set of skills in BridgeData V2. We define a skill to be a group of trajectories that require similar motions from the robot but may include different objects and start/end positions.

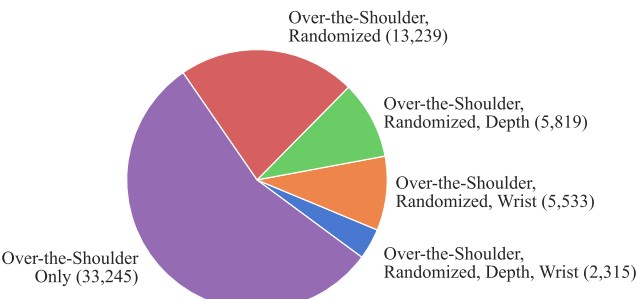

**Figure 6** Breakdown of the entire dataset, including the autonomously collected data, by what camera views are included. "Over-the-shoulder" refers to the primary fixed camera, and "randomized" refers to the two alternative camera views that are randomized by the data collectors every 50 trajectories. "Depth", when present, is from the same perspective as the primary fixed camera. "Wrist" refers to the wide-angle wrist-mounted camera. More cameras were added to the hardware setup throughout data collection, so the majority of the data only includes the primary fixed camera view, and very little data currently includes all 4 views. However, now that the hardware is in place, more and more data will include all 4 views as the dataset continues to grow.

## A   Data Statistics

We provide a full list of the skills in our dataset (following our definition of skill in Section 3.3) in Table 5. We provide a breakdown of which portions of the dataset include which sensors in Figure 6.

## B   Learning Method Implementation Details

Below we list relevant implementation details for each method. The complete set of evaluation tasks is shown in Figure 7.

All the goal-conditioned methods take in both an observation and goal. During training, the goal associated with an observation is selected by uniformly sampling an observation from the future timesteps in the trajectory.

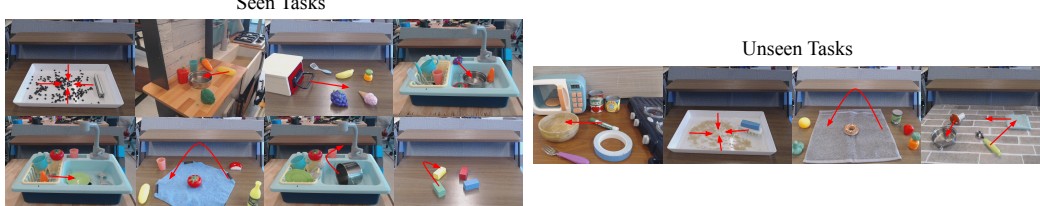

**Figure 7** The complete set of evaluation tasks. The seen tasks are (clockwise): sweeping beans into a pile, putting a corn cob in a pot, opening a drawer, putting an eggplant in a pot, stacking a block, flipping a pot upright, folding a cloth, and putting a carrot on a plate. The unseen tasks are (left to right): putting a marker in a bowl, sweeping rice into a pile, folding a thick cloth, putting a mushroom in a pot, putting a spoon on a cloth, and wiping the table with a cloth.

## B.1  Goal-conditioned behavior cloning

The observation and goal are stacked channel-wise before being passed into a ResNet-34 image encoder. This image encoding is passed through 3 256-unit fully connected layers to predict the robot action. We augment the observation and goal with random crops, random resizing, and color jitter. We use the Adam optimizer [51] with a learning rate of 3e-4. We use a linear warmup schedule with 2000 steps.

## B.2  Diffusion goal-conditioned behavior cloning

In our implementation, we adopt the behavior cloning with diffusion strategy from the IDQL [38]. However, we do not learn a value function. The observation and goal are stacked channel-wise before being passed into a ResNet-34 image encoder. This image encoding is used to condition a diffusion process that models the action distribution. We use the DDPM (Denoising Diffusion Probabilistic Models) style objective as introduced by Ho et al. [42]. We augment the observation and goal with random crops, random resizing, and color jitter. We use the Adam optimizer [51] with a learning rate of 3e-4. We use a linear warmup schedule with 2000 steps.

## B.3  Action Chunking with Transformers

We use the same ACT hyperparameters as the original paper [17] except for the chunk size. The original ACT has chunk size 100 to accommodate for high-frequency control (50Hz) and long trajectories (1000 steps), while in our case the control frequency is 5Hz and the trajectory is much shorter at around 50-100 steps. We therefore reduced the chunk size to 5 and noted better performance. In addition, ACT was originally proposed as a single task method; we modify ACT to make it goal-conditioned. The observation and goal are stacked channel-wise before being passed into a ResNet-18 image encoder. The ACT policy was trained on a consumer grade workstation with 1x Nvidia 2080Ti GPU for 3 days.

## B.4  Contrastive RL

Our implementation of contrastive RL mostly follows the design decisions described in [45] except for the following changes. First, given the observation and goal images, we feed them separately through a ResNet-34 encoder instead of a 3-layer CNN image encoder to get output encodings. Those image encodings then pass through two MLPs to get representations of the observation and the goal. Second, we share the ResNet-34 encoder between the value function and the policy. Intuitively, this encourages sharing of the causal information learned by the representations and speeds up learning. This ResNet backbone helps the method to consume large amounts of training data. Third, we increase the GCBC regularization coefficient to 0.2 to avoid sampling out-of-distribution actions and use the same batch size as other methods to make fair comparisons. Our contrastive RL objective retains the temporal-difference (TD) style used in [44]. We use the Adam optimizer [51] with a learning rate of 3e-4. We use a linear warmup schedule with 2000 steps.

| Task | BridgeData V1 + PTR | BridgeData V2 |
|------|:---:|:---:|
| Put marker in bowl[†] | 0.05 | 0.65 |
| Put mushroom in pot[‡] | 0.10 | 0.70 |
| **Average** | **0.08** | **0.70** |

[†] Unseen objects, unseen environment  [‡] Seen objects, unseen environment

**Table 6 (Dataset size and diversity ablation)** Comparison of the performance of GCBC trained on only BridgeData V1 + PTR (13k trajectories, 15 environments, 11 skills) and GCBC trained on BridgeData V2 (60k trajectories, 24 environments, 13 skills). The greater size and diversity of BridgeData V2 enables significantly better generalization to these unseen tasks.

### B.5   Language-conditioned behavior cloning

Our implementation of LCBC uses a ResNet-34 with FiLM conditioning. The language instruction is first encoded with a frozen MUSE encoder and passed through 2 fully connected layers. The image observation is then passed into the ResNet, which is conditioned on the language embedding using FiLM layers. FiLM layers are applied at the end of every ResNet block to condition on language throughout the network. Finally, the image encoding is passed into a fully connected network to predict the action. We use the Adam optimizer [51] with a learning rate of 3e-4. We use a linear warmup schedule with 2000 steps.

### B.6   RT-1

We use the same hyper-parameters as the original RT-1 paper [7], except for increasing the sequence length of the transformer from 6 to 15 to accommodate for the longer episode length. We scale each action dimension to range -1 and 1 and use a vocabulary size of 256 to tokenize the actions.

## C   Hardware Setup

We use an Intel RealSense D435 RGBD camera as a fixed over-the-shoulder camera view and two Logitech C920 webcams to capture alternative camera views that are randomized during data collection. For the wrist camera, we designed a custom 3D printed mount to attach a Raspberry Pi camera module to the gripper. We used a Meta Quest 2 VR headset to teleoperate the robot.

## D   Comparison to the Original Bridge Data

In Table 6, we compared the performance of GCBC trained on the combination of the original Bridge Data and data collected for PTR [36], with the performance of GCBC trained on all of BridgeData V2. We find that the additional data released as part of this paper significantly improves performance on unseen pick-and-place tasks.

