# OpenReview forum: "BridgeData V2: A Dataset for Robot Learning at Scale"
_robot-learning.org/CoRL/2023/Conference — CoRL 2023 Poster_

### Official Review · Reviewer_dwgC · 2023-07-16

**Confidence:** 4
**Originality:** Good
**Technical Quality:** Good
**Clarity Of Presentation:** Very Good
**Impact:** 4

**Recommendation:**

Weak Accept: I recommend accepting the paper, but will not argue for my recommendation if the majority of other reviewers have a different opinion.

**Review:**

This paper itself is clearly well written and no doubt would be useful for the robot learning community particularly for the ones focusing on large-scale pre-training methods. Here are some other concerns and questions which hopefully could further improve the paper.

1.	Transferability of different robot. All trajectories collected seem to be on the same WidowX 250 robot arm, which from my own experience, is not a common robot used in many labs. I am wondering how the policy learned through such robot arm can transfer to other robot arms such as Franka or xArm?
2.	Details on language prompts. The quality and graduality of language prompts will heavily influence the generalisation of the dataset. I am wondering how detailed, or the length of language prompts are for each task? E.g. labelling a task as: make a coffee and first grasp a cup to the coffee machine, then press the button… would lead to very different results.


**Quality Of The Limitations Section:**

Limitations are addressed clearly

**Questions For Rebuttal:**

See review.

**Robotics Focus:**

Sufficient demonstration on hardware

**Summary Of Paper:**

This paper presents a new dataset BridgeDataV2, by the name, is clearly an extended version of the original BridgeData. It contains significantly more trajectories with more environments all collected through human experts compared to the original dataset. As an aim for providing a good source for training multi-task foundational robot policy, the paper further evaluates the dataset with multiple popular multi-task offline RL and language-conditioned methods. Results show that the policies trained on this dataset can offer good generalisation on novel objects and new environments.

**Summary Of Recommendation:**

Proposes a useful dataset that would be useful for the robotics community; evaluations are clean and complete

---

> ### Author Response · Authors · 2023-08-15
> **Follow up**
>
> Today is the last day of the rebuttal period. Please let us know if our response has addressed your concerns, or if we can provide any further clarification. We would be happy to discuss in more detail. Thank you!

---

### Official Review · Reviewer_xJRp · 2023-07-18

**Confidence:** 5
**Originality:** Excellent
**Technical Quality:** Excellent
**Clarity Of Presentation:** Very Good
**Impact:** 4

**Recommendation:**

Strong Accept: I recommend accepting the paper and will argue for my recommendation even if other reviewers hold a different opinion.

**Review:**

## Strengths

 - **A Robot Dataset Paper**: Datasets are very important but remain understudied within robot learning. I think this project made an important contribution in investigating scaling laws of dataset size, diversity, and compute resources (e.g, parameter count) while keeping in mind challenges specific to robotics (hardware accessibility, diverse algorithm/policy/task specification formulations).
 - **Not just another rigid pick and place dataset**: Includes deformable and granular manipulation.
 - **Careful multi-task environment design**: with multiple possible tasks given the same visual observation.
 - A small point but every multi-task paper should define clearly what they mean when using overloaded terms (e.g., tasks and skills), like this one.


## Weaknesses

**Unsystematic Evaluation**: Different tasks have different caveats and challenges (e.g., as explained in L234-L237). This is fine, and explaining this to readers is important. However, each table chose a different set of tasks, while not containing consistent coverage of all skills/objects. This makes the evaluation section confusing, because each new table requires readers to consider that table's tasks' caveats while looking at the numbers.
Further, without knowing exactly the degree to which these caveats affect the numbers, it makes it difficult for readers to extract takeaways from the reported numbers.

**Breath over Depth**:
The project's main motivation is to investigate how a diverse dataset could facility a diverse group of people with access to different resources to study scalable robot learning.
Given this motivation, I think their evaluation design choice of breath over depth (many algorithms under different generalization axes for environments, objects, tasks) is more aligned with the motivation than the reverse.
Breath over depth is not a weakness, but my main concern is with respect to *confidence*. The number of tasks and the number of episodes per task (10 trials per task) evaluated do not lead to high confidence in reported numbers and, therefore, generalizable conclusions for others to learn from. For instance, I think the scaling law plot (Fig 5, left) is extremely exciting to see. However, this plot was generated using one task, which is to move a spoon.
 - **Selective Breath over Depth**: The authors demonstrated that multiple policy learning formulations could learn from the data. However, to investigate research questions 2, 3, and 4 (in the beginning of section 5), it may not have been necessary to run costly real world evaluations on all 6 approaches for all research questions, allowing more trials to be allocated to each approach (e.g., 20 trials in [18]).

**Non-foundational robot skills**:
Compared to the zipping/unzipping task, the paper and supplementary website contains many more visualizations and results for other tasks.
The dataset also contains many more trajectories for other tasks.
  - **Visualizations**: For this non-standard task, it will be helpful to also include visualizations of the zipping/unzipping task. From the supplementary website, the `traj_links.csv` only include 100 examples of unzip, and only include the first and last frame, but no "zip" task.
  - **Usage of Non-foundational skills**: While less foundational than object rearrangement, more unique skills like such as zipping and unzipped (I do like this task!) are interesting to study within the context of developing a foundation multi-task model for robotics (and not just a foundation pick and place models, for which we already have good data-efficient solutions).
    -  Exactly how many zipping/unzipping trajectories are included in this dataset? Were they used for any experiments? Do such a small amount of trajectories for a task add value to the dataset? Including a discussion of these questions will be interesting for future work to understand how they can contribute to BridgeDataV2 (e.g., by adding less studied skills).

**Quality Of The Limitations Section:**

Limitations are addressed clearly

**Questions For Rebuttal:**

See questions and concerns in weaknesses above.

**Robotics Focus:**

Sufficient demonstration on hardware

**Summary Of Paper:**

With the motivation of facilitating large-scale robot learning research, the paper introduced a real-world crowdsourced-language-labelled dataset, on an accessible hardware platform, with diverse environments, objects, and skills.
They demonstrated that a diverse set of robot policy learning formulations and a different lab can make use of this data.
They also provided some empirical evidence that performance scales with dataset size and compute, which motivates future cross-institutional collaboration on large scale robot dataset collection.

**Summary Of Recommendation:**

I think the project's motivation is spot on and its contributions could be extremely valuable to the community.
Investigating datasets are important, challenging to study deeply and broadly, and crucially understudied in the robot learning community, and the field would benefit a lot from more projects in this direction.
However, I do not think the project's ambitious research questions were sufficiently empirically backed up by their experiments.
For a weak accept, I would like to see a restructuring of the evaluation section (addressing my **Unsystematic Evaluation** point above).
For a strong accept, I would also like to see my statistical confidence concerns due to **breath-over-depth** evaluation design addressed.
I think the paper can be educational after addressing my concern regarding **usage of non-foundational robot skills**, but that alone will not be enough to change my recommendation.

**Update**
Evaluation and statistical significance concerns addressed. `Strong Accept`.

---

### Official Review · Reviewer_SSre · 2023-07-19

**Confidence:** 4
**Originality:** Good
**Technical Quality:** Good
**Clarity Of Presentation:** Very Good
**Impact:** 3

**Recommendation:**

Weak Reject: I recommend rejecting the paper, but will not argue for my recommendation if the majority of other reviewers have a different opinion.

**Review:**

Standard benchmarking of learning algorithms is a worthy goal, and this paper takes a step towards better robotic benchmarks. It is evident that much time and effort were poured into the creation of the described dataset, both in planning which tasks and environments to include and in designing the data collection protocol.

The paper is easy to read and follow, and its contributions are stated clearly and concisely. The inclusion of benchmarking experiments on the collected dataset is a nice addition, and helps with evaluating the usability of the dataset.

If the dataset is publicly released and continuously supported by the authors, it could be a useful addition to the toolbox of benchmarks available to robotic learning researchers.

Nonetheless, I am concerned that the claims of generality and diversity made by the authors of this paper may be overstated.

In terms of diversity, as mentioned in the limitations section, all tasks require relatively simple manipulation with no dynamic movements or contact-rich interactions between objects. While some of these manipulation tasks are still quite challenging, many of them have been solved individually by approaches that do not require such a large dataset of previous interactions. In addition, the toy kitchen, sink and tabletop environments which constitute the majority of the dataset are not as diverse and different from each other as the authors describe.

Regrading generality, the entire dataset is collected using a single model of robot, which may not be useful or applicable to many tasks outside of the toy domains chosen for the dataset, due to its limited reach, degrees of freedom and carrying capacity. In addition, while this robot may be readily available in North America, it may not be as easy to assemble all the required components to replicate a setup similar to the one presented in this paper in other parts of the world.  I believe that datasets for generalizing robot learning should be applicable to whichever setup a research lab already has installed, or at the very least, utilize a more commonly used or more generally useful robot arm.

Finally, it seems like the dataset is not yet complete - as described in the appendix (and evident in the choice of observations for the methods benchmarked), most trajectories do not have depth images attached, and only use an RGB observation from the shoulder camera. Moreover, as the dataset was collected in a single lab on a single setup, the diversity of lighting conditions and other environment variables may be too small to generalize well. A larger comparison study of the setup in other labs would be more convincing of the usefulness of the dataset.

**Quality Of The Limitations Section:**

Limitations are addressed clearly

**Questions For Rebuttal:**

While it is true that most methods exhibit non-zero success on generalization tasks, it is somewhat hard to assess the quality of the dataset from the standalone performance of the given methods. How do the various baseline methods perform on other, previous datasets?

What tuning and setup challenges, if any, did the other institution (Lab 2) encounter while replicating the environment and evaluating the methods to obtain the results reported in Table 5?

(On a side note, the text only mentions two methods were evaluated in Lab 2, while Table 5 reports results for all methods).

**Robotics Focus:**

Sufficient demonstration on hardware

**Summary Of Paper:**

This paper proposes an extension of the Bridge Dataset (Ebert et al. 2021), with over 50K trajectories from a variety of toy tasks performed by a WidowX 250 arm. The dataset, comprising mostly of human demonstrations and some heuristically collected pick-and-place data, includes image observations, robot actions taken in the environment and human-annotated descriptions of the tasks performed in each trajectory. In addition, the authors provide evaluations of offline learning methods trained on the collected data.

**Summary Of Recommendation:**

While the proposed dataset could undoubtedly be useful to some researchers benchmarking offline learning or tabletop manipulation algorithms, I find its contribution to the robot learning community limited due to the choice of robot and tasks. A larger, more diverse dataset or an extended comparative study of applying the dataset in other labs may be a convincing addition which could greatly improve this paper.

---

### Official Review · Reviewer_AXMo · 2023-07-20

**Confidence:** 4
**Originality:** Good
**Technical Quality:** Good
**Clarity Of Presentation:** Good
**Impact:** 3

**Recommendation:**

Weak Accept: I recommend accepting the paper, but will not argue for my recommendation if the majority of other reviewers have a different opinion.

**Review:**

Strengths:
  1. Compared to other real-world datasets, BridgeData V2 includes more skills and environments.
  2. The hardware setup is low-cost and friendly to many researchers.
  3. The experiments across labs are interesting, demonstrating the diversity and generalization of the proposed dataset.

Weaknesses:
  1. The authors should provide more details about the experiments across labs. What is the difference between these labs? Are manipulated objects,  background, light, or hardware setup different? These details will make the experimental results more convincing.
  2. The experimental results in Table. 2, Table. 3 and Table. 4 are confusing. Different types of methods are evaluated for different tasks in this paper. Additionally, in Table. 3, the methods perform much better in unseen environments than in seen environments. RT-1 achieves nearly 1.8x performance improvement, which is weird. The authors should analyze the results with more details.
  3. The setup of camera views (number of views, provided depth or not) is not the same for all trajectories, which will be burdensome for future researchers in preprocessing data and designing methods.

**Quality Of The Limitations Section:**

Limitations are addressed clearly

**Questions For Rebuttal:**

As addressed in Weaknesses, the authors should provide more details about:
  1. What is the difference between these labs? I think the authors could provide a figure to visualize the difference between labs or a table to list what is different exactly.
  2. The analysis of experimental results should focus on the datatset, rather than simply comparing the baselines, since the main contribution is the dataset. The authors should also explain why the methods perform much better in unseen environments than in seen environments in Table. 3.
  3. The authors should do some experiments to study how the diversity of skills Impact the generalization.

**Robotics Focus:**

Sufficient demonstration on hardware

**Summary Of Paper:**

This paper introduces a larger dataset of robotic manipulation named BridgeData V2, which is collected in the real world with natural language labels. BridgeData V2 is suitable for both IL and  offline RL methods. The system setup is low-cost, making it accessible for many researchers.

**Summary Of Recommendation:**

As stated in weaknesses and questions, my final rating is Weak Reject, but I am glad to hear feedbacks from the authors during the rebuttal.

New: -> Weak Accept

---

> ### Author Response · Authors · 2023-08-15
> **Follow up**
>
> Today is the last day of the rebuttal period. Please let us know if our response has addressed your concerns, or if we can provide any further clarification. We would be happy to discuss in more detail. Thank you!

---

### Author Response · Authors · 2023-08-08
**Rebuttal**

We thank the reviewers for their time and constructive feedback. As an overall comment, we want to emphasize that the goal of our paper is to provide a dataset that is useful to researchers interested in studying broad generalization in robot learning. There are no existing publicly available, language-annotated robot datasets of comparable size and diversity (see Table 1). The closest is BC-Z with 26k trajectories collected on a single table with a proprietary robot, while BridgeData V2 contains 54k trajectories collected in many environments with a robot that is available to buy. Our experiments demonstrate that our dataset enables learning a single multi-task open vocabulary policy capable of generalizing to new objects, environments, and even across institutions. Very little prior work studies generalization at this scale. The few existing public datasets of similar scale mostly focus on simple language-conditioned pick-and-place tasks [[RT-1](https://arxiv.org/abs/2212.06817), [BC-Z](https://arxiv.org/abs/2202.02005)], while we evaluate many more complex skills, like granular sweeping or folding cloth (see Tables 2, 3, and 4 and our [website](https://bridgedata-v2.github.io/) for videos). More importantly, the primary goal of these prior works is not to provide a useful resource to the community. They use proprietary robots and only evaluate at a single institution, whereas our dataset uses a robot that is cheap and available for purchase as well as a variety of environments (to facilitate generalization to new settings) many of which (e.g., toy kitchens) can be purchased for users who wish to reproduce our results.

We have responded to the specific concerns of each reviewer below. Please see the attached PDF in each reply for additional results that address the issues raised.

---

### Decision · Program_Chairs · 2023-08-30

**Decision:**

Accept (Poster)

**Comment:**

Summary of the Paper:

The paper introduces BridgeData V2, an expanded robotic manipulation dataset containing real-world data with natural language labels. This dataset, suitable for imitation learning and offline reinforcement learning, promises a more diverse set of trajectories and environments than its predecessor. It is primarily collected using the WidowX 250 robot arm and covers a variety of toy tasks. The cost-effective hardware setup makes it appealing to a wide range of researchers. Furthermore, the paper evaluates the dataset with a variety of multi-task offline RL and language-conditioned methods, aiming to showcase its potential for training multi-task foundational robot policies.

Strengths:

- Rich and Accessible Dataset: BridgeData V2 offers more skills and environments compared to other datasets, making it a valuable resource for robotic research.
- Cost-Effective Hardware Setup: The low-cost hardware setup is lauded by reviewers for making the research more accessible to many.
- Diversity in Manipulation: The dataset includes not just rigid pick and place tasks but also deformable and granular manipulation which is a noted positive.
- Comprehensive Evaluation: The dataset's evaluation using multiple popular offline RL methods provides insight into its usability and potential.

Weaknesses:

- Lack of Detailed Information: Several reviewers pointed out the need for more detailed information on various aspects:
- Experimental setups across different labs.
- Inconsistencies and confusing results in tables (especially Tables 2, 3, and 4).
- Variability in camera setup for different trajectories.
- Limited Diversity and Generality: Concerns were raised about the dataset's diversity, as all tasks seem to be simple manipulations, lacking dynamic movements or interactions between objects.
- Specific Robot Used: The dataset's reliance on the WidowX 250 robot arm might limit its applicability, as this model is not common in many labs. Questions arose regarding the transferability of policies learned on this robot to other models.
- Inconsistency in Evaluation: The presentation of results across different tables made the evaluation section confusing for readers. Moreover, the choice to go breadth-over-depth in an evaluation led to concerns about the statistical confidence in reported numbers.
- Language Prompts Concern: The quality, graduality, and specificity of language prompts can influence the dataset's generalization, but details on this aspect were seen as lacking.

Rebuttal and Discussion:

The authors responded diligently to the questions raised by the reviewers and managed to address several of them. However, not all concerns have been fully resolved, and further revisions are still required.

Overall Recommendation:

Given the feedback from all reviewers, rebuttal, and discussion, the consensus leans towards "Weak Accept" with certain reservations. The dataset, no doubt, adds value to the community, but there are areas in the paper that need revision, clarification, or additional detail. Addressing the mentioned weaknesses, especially in terms of the detailed information and clarifying confusing results, could elevate the paper's impact and acceptance within the community.